# Through the big top: An exploratory study of circus-based artistic knowledge translation in rural healthcare services, Québec, Canada

**Julie Théberge**[1,2,3,4☯], **Mélanie Ann Smithman**[5‡], **Catherine Turgeon-Pelchat**[2‡], **Fatoumata Korika Tounkara**[2‡], **Véronique Richard**[4,6‡], **Patrice Aubertin**[4‡], **Patrick Léonard**[7‡], **Hassane Alami**[8‡], **Diane Singhroy**[9‡], **Richard Fleet**[1,2,3☯]*

1 Laval University, Québec, Canada, 2 Research Chair in Emergency Medicine Université-CISSS-CA, Lévis, Canada, 3 VITAM Centre de recherche en santé durable, Québec, Canada, 4 Centre de recherche, d'innovation et de transfert en arts du cirque, Montréal, Canada, 5 Unity Health Toronto, University of Toronto, Toronto, Canada, 6 University of Queensland, Brisbane, Australia, 7 Sept Doigts de la main, Montréal, Canada, 8 University of Oxford, London, United Kingdom, 9 McGill University, Montréal, Canada

☯ These authors contributed equally to this work.
‡ PL, HA and DS also contributed equally to this work.
* fleetrick1@gmail.com

## Abstract

### Background

The conventional methods and strategies used for knowledge translation (KT) in academic research often fall short in effectively reaching stakeholders, such as citizens, practitioners, and decision makers, especially concerning complex healthcare issues. In response, a growing number of scholars have been embracing arts-based knowledge translation (ABKT) to target a more diverse audience with varying backgrounds and expectations. Despite the increased interest, utilization, and literature on arts-based knowledge translation over the past three decades, no studies have directly compared traditional knowledge translation with arts-based knowledge translation methods. Thus, our study aimed to evaluate and compare the impact of an arts-based knowledge translation intervention–a circus show–with two traditional knowledge translation interventions (webinar and research report) in terms of awareness, accessibility, engagement, advocacy/policy influence, and enjoyment.

### Methods

To conduct this exploratory convergent mixed method study, we randomly assigned 162 participants to one of the three interventions. All three knowledge translation methods were used to translate the same research project: "Rural Emergency 360: Mobilization of decision-makers, healthcare professionals, patients, and citizens to improve healthcare and services in Quebec's rural emergency departments (UR360)."

### Results

The findings revealed that the circus show outperformed the webinar and research report in terms of accessibility and enjoyment, while being equally effective in raising awareness,

**Data Availability Statement:** All relevant data are within the manuscript and its Supporting Information files.

**Funding:** The project received financial support from the AUDACE fund of the FRQSC-Fonds de Recherche du Quebec/Society and Culture (2021-AUDC-286902) allocated to Fleet and Aubertin.

**Competing interests:** The authors have declared that no competing interests exist.

increasing engagement, and influencing advocacy/policy. Each intervention strategy demonstrates its unique array of strengths and weaknesses, with the circus show catering to a diverse audience, while the webinar and research report target more informed participants. These outcomes underscore the innovative and inclusive attributes of Arts-Based Knowledge translation, showcasing its capacity to facilitate researchers' engagement with a wider array of stakeholders across diverse contexts.

## Conclusion

As a relevant first step and a complementary asset, arts-based knowledge translation holds immense potential in increasing awareness and mobilization around crucial health issues.

## Introduction

Researchers face significant challenges disseminating their research to a wide range of stakeholders, such as citizens, practitioners, and decision makers [1–3]. In the context of Canadian healthcare, knowledge translation is defined as a dynamic and iterative process that includes synthesis, dissemination, exchange and ethically-sound application of knowledge to provide Canadians more effective healthcare services and products and strengthen the healthcare system [4]. It aims to move important health research evidence into the hands of people and organizations who can put it into practice. Knowledge translation (KT) methods such as scientific journal articles, research reports, policy briefs, presentations at scientific events and webinars are well entrenched in the culture of healthcare research [5]. Although essential to scientific literacy, traditional knowledge translation often fall short of influencing stakeholders and driving meaningful action around complex social issues (ex: health inequalities, mental health) [6]. Brownson et al. [7] note that it takes an average of 17 years for 14% of health research to be useful to patients, populations and/or healthcare professionals. Significant amounts of financial, technological, and human resources are invested to conduct valuable research. To ensure a fair return on investment and ultimately improve population health, developing appropriate knowledge dissemination method and strategies are crucial [8].

KT initiatives are complex and multifaceted endeavors that are constrained by the rigidity of academic disciplines [9–12]. The emergence of community-based research and the importance of involving patients, communities, and the general public forces us to rethink the classical modes of KT. The COVID-19 crisis has underscored the need for innovative knowledge translation strategies capable of reaching a broader audience, including the general population, which conventional knowledge translation methods alone cannot attain. Consequently, researchers should reassess their approaches to translating science into accessible knowledge [13].

To optimize the impact of research findings, a growing number of scholars are exploring arts-based knowledge translation. Arts-based knowledge translation is part of the arts-based research (ABR) field [14–16]. This methodology harnesses the power of visual and literary arts, performing arts, design and crafts by integrating them into community via cultural festivals, fairs and events, online or in person [17]. It aims to communicate research findings to target audiences and facilitate the co-production of knowledge and social change [18]. Arts-based knowledge translation crosses cultures, individual and collective experiences and representations, professional domains, and identities. They are a means to navigate complex and dynamic systems [19].

The interest, use and literature on arts-based knowledge translation and arts-based research have increased in the last thirty years among researchers, artists and knowledge brokers

particularly in the healthcare sector (Art-Based Health Research, ABHR) to address and investigate a wide range of health issues and populations [20–23]. Arts-based knowledge translation in health informs clinical practice and public understanding of key health issues [24]. It acts as a catalyst for dialogue between diverse stakeholders and is very much in line with participatory and community-based research [17]—a priority of major international granting agencies [25]. Arts-based knowledge translation projects have thus been found to bridge the gaps between evidence and practice as well as knowledge and action in health issues [26].

Our team has previously explored the circus arts as an artistic means to increase awareness of the principles of creativity at major scientific conferences such as: Family Medicine Forum (FMF Quebec city, 2014), the Canadian Association of Emergency Physician (CAEP, Québec city 2016) [27] and partnered with medical associations such as Canadian Medical Association (CMA) and Association des Médecins d'Urgence du Québec (AMUQ) (Québec Association of Emergency Physicians) to promote rural healthcare services. In these conferences, keynote lectures were supported by circus acts and illustrated the potential of simple artistic performances in highlighting key messages with emotion. Over 4000 physicians and researchers were reached in person through these arts-based knowledge translation initiatives. While conference evaluations were highly positive, no scientific evaluation of the impact of these initiatives was planned and conducted.

Circus is well established in the province of Quebec (Canada), and Montreal is considered by many as the circus capital of the world, home to the Cirque du Soleil, Les 7 Doigts and the National Circus school [28]. Coincidentally, members of our research team were involved in both circus arts and rural healthcare research. Despite widespread efforts to disseminate the results of a decade of research demonstrating the disparities between rural and urban access to care through traditional means (e.g., scientific publications, presentations, social and traditional media), the impact on mobilizing stakeholders towards change remains limited. Similarly, despite expert recommendations, studies comparing traditional knowledge translation with arts-based knowledge translation methods are scarce [18, 20, 23]. In this context, we inquire whether the circus arts can aid researchers in bridging the gaps between evidence and practice, and between knowledge and action, concerning health issues.

This study seeks to conduct a comparative analysis of an Arts-Based Knowledge Translation circus show against two conventional knowledge translation methods (a scientific study report and a webinar) to evaluate their impact on perceptions of accessibility, engagement, awareness, advocacy/policy change, and enjoyment.

Our primary hypothesis posited that the circus show would outperform traditional knowledge translation methods across all variables. However, we also anticipated that all interventions, including the circus show, would play a role in raising awareness of critical rural healthcare challenges and the proposed solutions for improvement. The information presented in the interventions was structured and assessed based on four main themes: 1) governance; 2) organization of care and services; 3) professional practice; and 4) access to resources.

## Materials and methods

### Design

We devised an exploratory convergent mixed-method study to enrich the scope of the research by leveraging the advantages of both qualitative and quantitative research while mitigating their respective limitations. This approach facilitated the comparison of data, allowing us to explore diverse, convergent, complementary, or divergent findings [29]. The integration of data followed two distinct intents: matching intents (collecting data to explore the relationship

between the two) and expanding intents (using data to gain a broader, yet overlapping under-standing of the phenomenon) [30].

## Procedures

The entire study spanned from January 2019 to August 2022, encompassing various phases, such as design and preparation (2019), show creation (2020), data collection (2021), and analy-sis (2022). Data collection occurred between April and May of 2021 and was carried out in the province of Québec, Canada. The KT interventions were scheduled during the same week, and due to COVID-19 pandemic restrictions, participants were engaged virtually. At the start of the week, all participants received a comprehensive email containing their randomly assigned intervention, along with links to the pre-intervention questionnaire and their respective inter-vention materials. The report and webinar groups were given four days to review their inter-vention content. In contrast, participants attending the circus show were requested to join the broadcast on the evening of the fifth day.

On the fourth day, the post-intervention questionnaire was dispatched via email to partici-pants in the report and webinar groups, which also served as a reminder to complete the pre-intervention questionnaire if necessary. For the circus show group, the post-intervention ques-tionnaire was sent immediately after the show. Participants who volunteered to partake in the discussion groups were provided with the relevant links and instructions (date and time) to join the discussions.

The research was conducted in partnership with Montréal based circus company "Les 7 Doigts" and the Research Center of the Montreal National Circus School (CRITAC).

## Recruitment and randomization

A call for voluntary participation was initiated in April 2021, inviting individuals from both rural and urban areas, healthcare professionals, researchers, and decision-makers associated with the UR360 study. To bolster recruitment efforts, we extended invitations to our network of partners and stakeholders with whom we had established collaborations in Quebec since 2012. The recruitment process was constrained to a limited timeframe of three weeks. Conse-quently, we employed a snowball method to facilitate completion. During this period, we sent personal and organizational invitations via email, and Facebook posts were sponsored and shared by all partners and stakeholders. Eventually, 162 participants registered and provided consent to take part in the study (Fig 1).

Participant eligibility criteria were: 1) 18 years old or older; 2) fluent in French; and 3) had access to technology and an internet connection. Participants who did not meet these require-ments and failed to provide complete contact information during the recruitment phase were excluded from the study. The team's statistician concluded that due to the absence of any com-parable theoretical study to estimate effect sizes, no meaningful statistical power calculation was possible.

Furthermore, participants were recruited through a convenience sampling method and ran-domized across interventional groups. Using the RAND function on Microsoft Excel, a ran-dom sequence was generated to assign participants to one of the three intervention groups in a 1:1:1 ratio.

## Variables

The effect of the three knowledge translation interventions were compared on five core vari-ables based on Kukkonen's Arts-Based Knowledge Translation Planning Framework [31]. For this study, we concentrated our efforts on four brokering goals as variables: 1) **accessibility** to

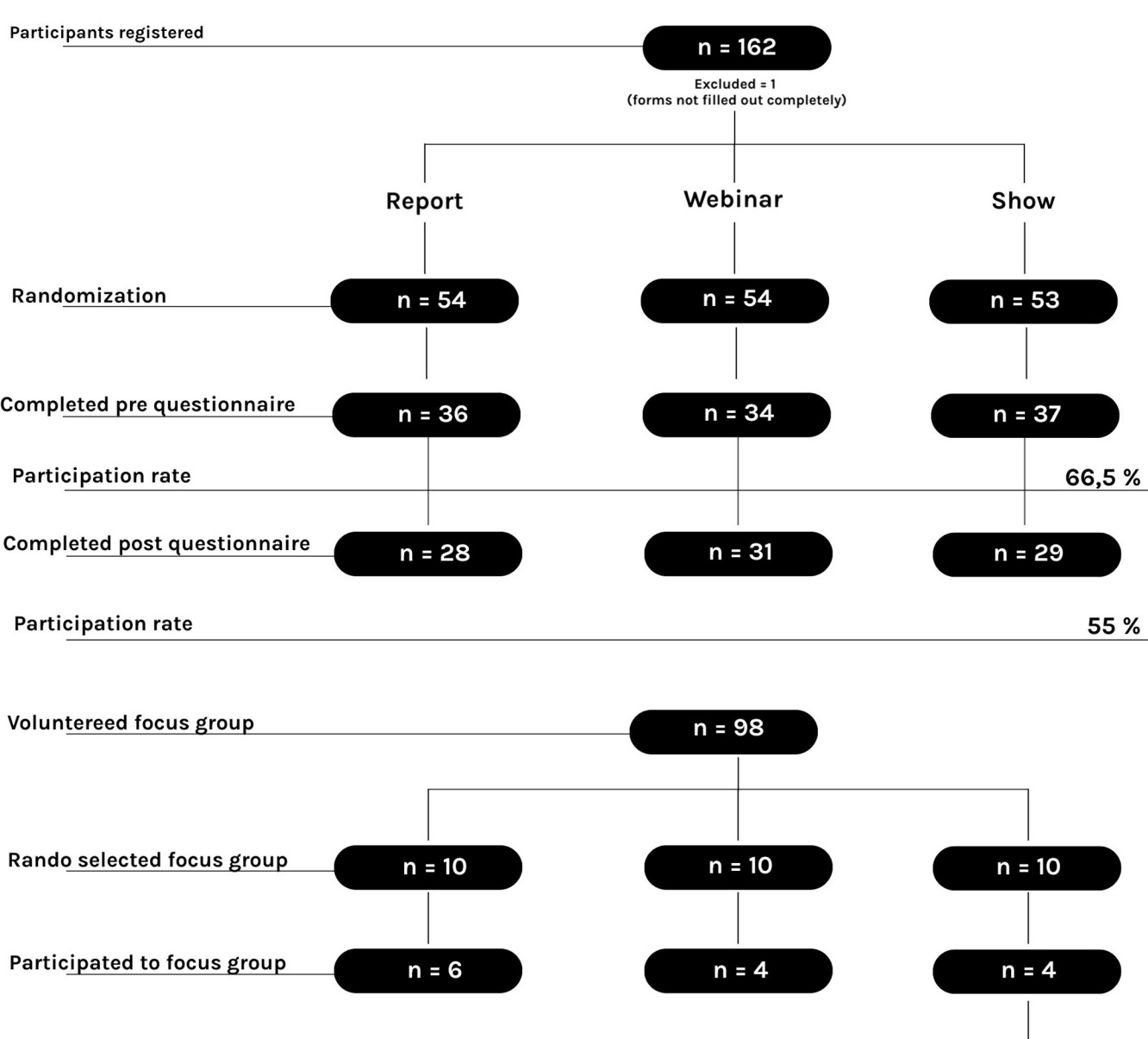

**Fig 1. Randomization flowchart.**

scientific research; 2) **awareness** of the empirical evidence on rural health issues; 3) **engagement** with research through making it appeal to more of the senses; 4) **advocacy and policy influence**: using research to stimulate change. We added the variable "**enjoyment**" to investigate its relation to the other four variables and to the outcome of the knowledge translation tool [32]. Previous studies from the field of pedagogy [33] and psychology [34] suggest positive emotions and aesthetics favor learning and likelihood to engage in sustainable, prosocial behavior.

## Quantitative data collection

We developed a pre-intervention closed questionnaire comprising twenty questions, divided into six sections: 1) demographic information; 2) general use of knowledge translation methods; 3) satisfaction with existing knowledge translation tools; 4) professional and personal needs related to knowledge translation methods; 5) interest and knowledge in rural healthcare services; and 6) motivation to enhance healthcare in rural areas. The questionnaire employed a seven-point Likert scale ranging from 1 to 7, alongside multiple-choice and open-ended questions.

The post-intervention questionnaire consisted of twenty-eight questions, organized into five sections: 1) assessment of enjoyment with the randomly assigned knowledge translation tool; 2) identification of strengths and weaknesses of the knowledge translation tool; 3) understanding of rural healthcare services; 4) motivation to improve healthcare in rural areas; and 5) perception of the relationship between art and knowledge translation. Like the pre-intervention questionnaire, the post-intervention questions were rated on a 1 to 7 Likert scale and included open-ended questions. Both the pre- and post-intervention questionnaires could be completed within a time frame of less than twenty-five minutes.

## Qualitative data collection

To gain further insights into the open-ended questions from the questionnaires, we conducted five discussion groups and two semi-structured interviews with diverse stakeholders. These discussions and interviews involved volunteers from each intervention group and took place within two to three weeks after the intervention. The primary aim was to delve deeper into participants' perspectives on the content and format of their assigned knowledge translation method. Participants were encouraged to elaborate on their level of interest in and utilization of knowledge translation methods, their enjoyment of the intervention experience, and their understanding of issues related to rural healthcare and citizen mobilization.

The discussion groups lasted around 60 minutes each and were conducted virtually, adhering to the COVID-19 health guidelines in effect at the time. To preserve the accuracy of the data, all interviews and discussions were recorded and subsequently transcribed by a member of the research team. The research team members, guided by an interview guide, led the interviews and discussions. The questionnaires and interview guides were designed by the research team and were pre-tested in a pilot study conducted in La Malbaie (Quebec) in July 2020. The entire research was carried out in Quebec and was conducted in the French language. To ensure clarity in the article, the quotes were translated by our research team, supported by the DeepL application.

## Interventions

The content of the three knowledge translation methods described below focused mainly on the results of a mixed methods study conducted in Quebec rural emergency: "Rural emergency 360: Mobilization of decision makers, healthcare professionals, patients and citizens to improve healthcare and services in Quebec's rural emergency departments (UR360)" (FRQS 2016–2019)" [35]. In this study of 26 rural emergency departments (EDs), Fleet *et al.* documented rural versus urban inequities in access to services in rural EDs in Canada [36, 37] and documented adverse clinical outcomes following a stroke [38] and trauma [39] in rural vs urban areas in Canada. All three knowledge translation conditions summarized the later studies.

**Scientific report.** The report (not yet published) is 32 pages long with annexes (Fig 2). It was designed as a classic end of study scientific report for decision makers and key stakeholders with general health care backgrounds. It is presented in a PDF colored format [40] including figures, tables and quotes from interviews covering the entire UR360 study.

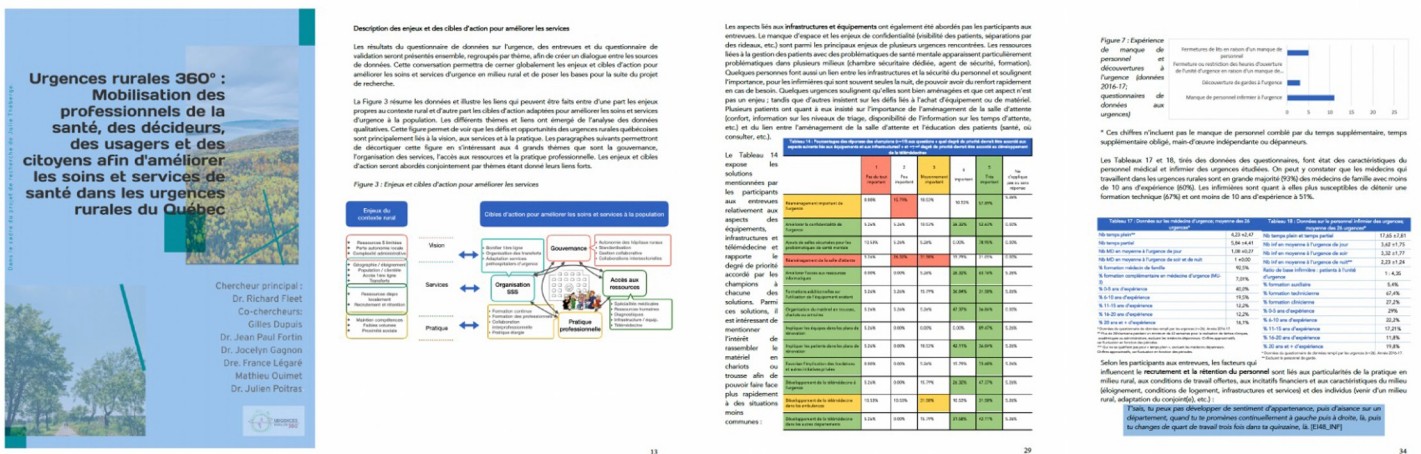

**Fig 2. Overview of pages from the study report (Intervention 1).** Reprinted from study report under a CCBY license, with permission from Richard Fleet, original copyright 2021.

**Webinar.** The webinar (Fig 3) consisted of a typical 45-minute PowerPoint presentation, featuring visual slides and audio clips from interviews. The principal investigator (R Fleet) delivered the presentation in his customary conference style, as instructed. The presentation was pre-recorded in a single take for convenience. Participants could access it at their leisure via a secret YouTube link [41].

**Circus show.** Lastly, the contemporary circus show (Fig 4) engaged four skilled circus artists to deliver a 55-minute stage performance, presenting an adaptation of the report through a fusion of circus techniques, theater, music, lighting, and video projection. The show aimed to strike a delicate balance between artistic prowess and accurate scientific information. Originally, the plan was to livestream the performance on a single evening through YouTube and

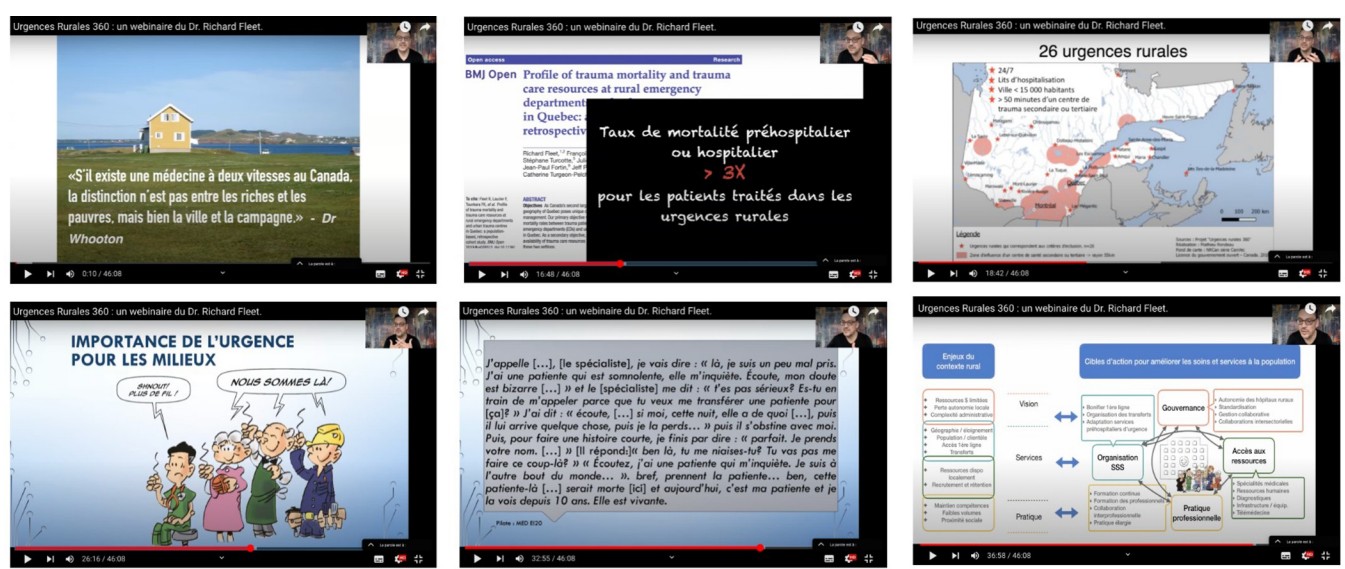

**Fig 3. Overview of slides from the webinar (Intervention2).** Reprinted from the PowerPoint presentation under a CCBY license, with permission from Richard Fleet, original copyright 2021.

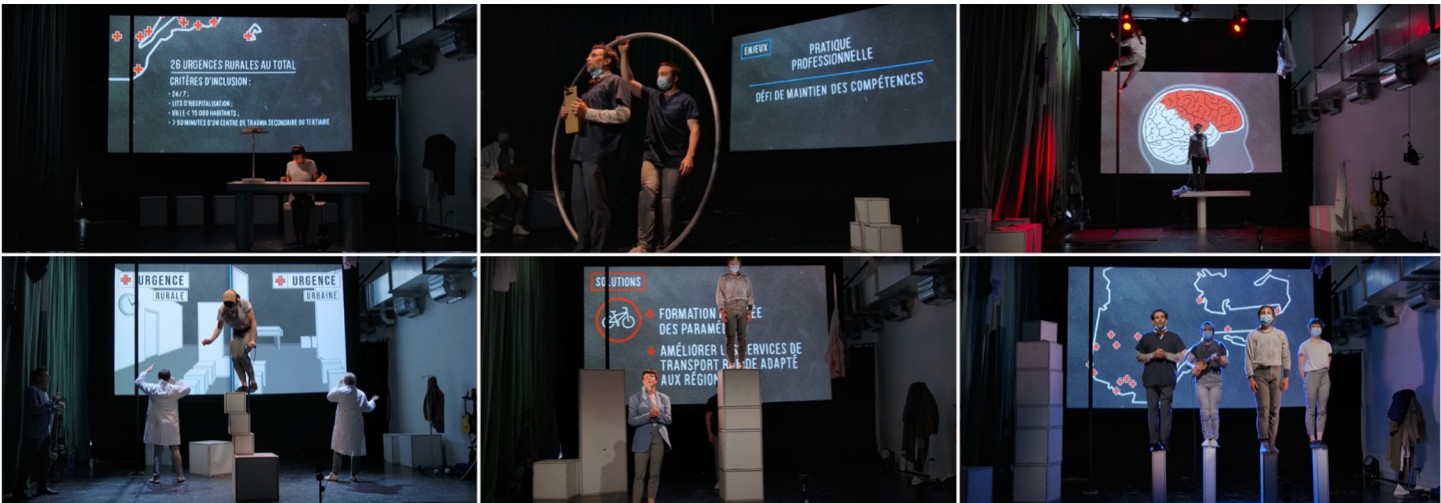

**Fig 4. Overview of scenes from the circus show (Intervention 3).** Reprinted from UR360 show under a CCBY license, with permission from Richard Fleet, original copyright 2021.

Facebook. However due to an unfortunate injury sustained by one of the artists, a recorded rehearsal was presented instead [42].

## Data analysis

Quantitative data analyses were performed using SAS Version 9.4 (SAS institute, Inc, Cary, NC, USA). Descriptive statistics were computed, and sociodemographic and data were presented as a percentage. When appropriate, Pearson's Chi-squared test [43] and Fisher's exact test [44] were used to compare these proportions. For qualitative data, we exported the open-ended question of the survey to Excel. We then proceeded in an inductive thematic analysis [45] using NVivo. The qualitative data was coded and summarized by research coordinator and first author (Theberge). To enhance reliability, three co-authors (Turgeon-Pelchat, Smithman, Richard) reviewed the codebook and discussed, challenged, and refined it. Throughout the investigation, team meetings were held to confirm accuracy and make modifications. The methodology was consistent with the Checklist for Reporting Results of Internet E-Surveys (CHERRIES) [46] and the Consolidated Criteria for Reporting Qualitative Research (COREQ) [47].

## Data integration

The integrated mix-methods data analysis strategy aims to enhance interpretability and meaningfulness by combining information from two different types of data. This strategy involves both enhancing the understanding of the data by combining them and corroborating findings in one data form with evidence from the other [29]. The process is depicted in Fig 5.

## Ethics approval

All recruitment occurred exclusively online through written submissions. Interested individuals accessed a hyperlink leading them to a detailed research information form. Participants provided consent by registering for the study. A designated contact person was readily available to address any additional queries. Participants were also assured of their right to terminate their participation at any point. Ethics approval was granted by the CISSS Chaudière-Appalaches ethics committee (2021–718).

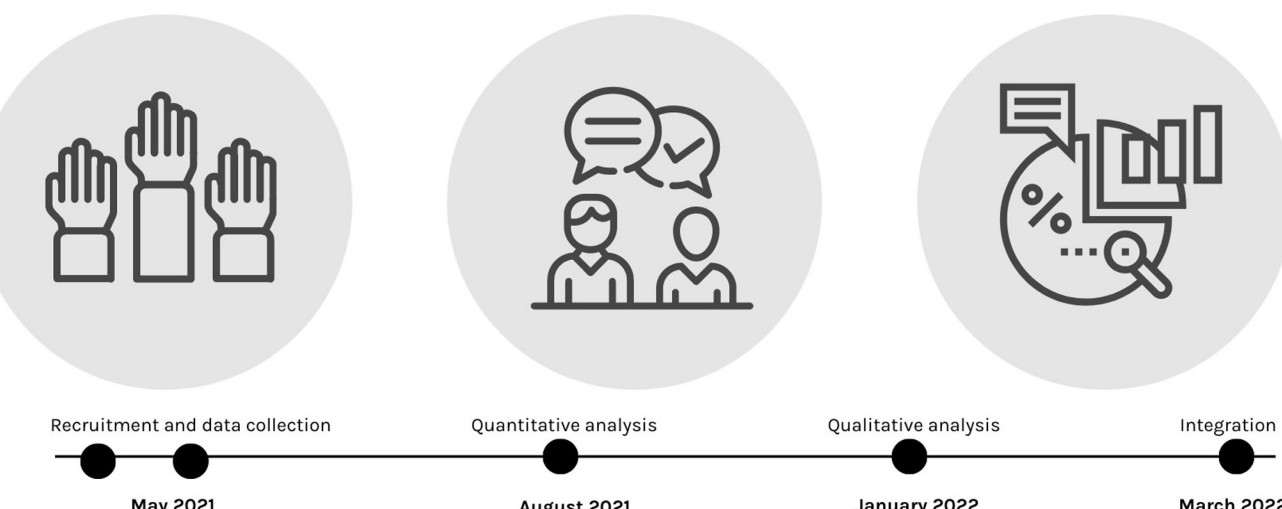

**Fig 5. Procedure flowchart.**

## Results

### Participants

Initially, a total of 162 individuals registered to participate in the study, comprising healthcare professionals, students, researchers, decision-makers, citizens, and artists (Table 1). Among them, 107 participants completed the pre-questionnaire, and ultimately, 88 individuals successfully completed the entire study. Out of the 107 participants, 76 (71%) identified as female, and

**Table 1. Baseline characteristics of participants according to randomization status before intervention.**

| VARIABLES / INTERVENTIONS | Webinar N = 34 | Report N = 36 | Show N = 37 | P-Value* |
|---|---|---|---|---|
| | n (%) | n (%) | n (%) | |
| **Age (years)** | | | | |
| $\leq$45 | 19 (55.9) | 21 (58.3) | 19 (51.4) | 0.831 |
| >45 | 15 (44.1) | 15 (41.7) | 18 (48.6) | |
| **Gender** | | | | |
| Female | 24 (70.9) | 26 (72.2) | 26 (70.3) | 0.981 |
| Male | 10 (29.1) | 10 (27.8) | 11 (29.7) | |
| **Profession** | | | | |
| Health workers, students, managers or researchers | 17 (50.0) | 17 (47.2) | 21 (56.8) | 0.704 |
| Citizens, artists or politicians | 17 (50.0) | 19 (52.8) | 16 (43.2) | |
| **Have ever lived in rural area** | | | | |
| Yes | 21 (61.8) | 18 (50.0) | 24 (64.9) | 0.399 |
| No | 13 (38.2) | 18 (50.0) | 13 (35.1) | |
| **Frequency of consultation of scientific communication tools** | | | | |
| Rarely | 4 (11.8) | 6 (16.7) | 7 (18.9) | 0.775 |
| Sometimes | 12 (35.3) | 10 (27.8) | 14 (37.8) | |
| Frequently | 18 (52.9) | 20 (55.5) | 16 (43.3) | |
| **Feels concerned by health issues in rural areas** | | | | |
| Yes | 28 (82.3) | 24 (66.7) | 28 (75.7) | 0.316 |
| No | 6 (17.7) | 12 (33.3) | 9 (24.3) | |
| **Knowledge of health care and services in rural areas** | | | | |
| Have no knowledge of health care and services in rural areas | 19 (55.9) | 18 (50.0) | 19 (51.4) | 0.876 |
| Have some knowledge of health care and services in rural areas | 15 (44.1) | 18 (50.0) | 18 (48.6) | |
| All P-Value were calculated with Pearson Chi-square test | | | | |

31 (29%) identified as male. Concerning age distribution, 59 participants (55%) were under 45 years old, while 48 participants (45%) were over 45 years old. Additionally, 63 participants (59%) reported having lived in a rural area at some point in their lives. A rural area was defined as a municipality with a population of less than 15,000 or located more than one hour away from a secondary or tertiary referral hospital. Regardless of their intervention groups, participants expressed concern about issues related to access to rural healthcare and self-assessed their knowledge about rural healthcare as moderate. Moreover, participants across all interventions demonstrated motivation to improve health services in rural areas. However, they felt inadequately equipped to actively contribute to the enhancement of such services. The study did not reveal any significant differences in terms of age, gender, professions, frequency of using scientific communication methods, or previous experience living in a rural area among the different interventions.

## Variables

Table 2 displays the results of the quantitative analysis. In general, a noteworthy distinction was observed between the circus show group and the other two groups concerning "Accessibility" (making scientific research accessible) and "Enjoyment" (global appreciation) (< 0,05.). There were no significant differences among the three groups for the other three variables: "Awareness" of the empirical evidence on rural health issues, "Engagement" with research through making it appeal to more of the senses and "Advocacy and policy influence" (using research to stimulate change). To complement and enhance our findings, we conducted

**Table 2. Participants appreciation according to randomization status after intervention.**

| VARIABLE / INTERVENTIONS | Webinar N = 31 | Report N = 28 | Show N = 29 | P-Value* |
|---|---|---|---|---|
| | n (%) | n (%) | n (%) | |
| **ACCESSIBILITY** | | | | |
| **Do you think this scientific communication tool can help make science accessible?** | | | | |
| Yes | 21 (70.0) | 19 (67.9) | 28 (100) | < 0.001 |
| No | 9 (30.0) | 9 (32.1) | 0 | |
| **ENGAGEMENT** | | | | |
| **Did the scientific communication tool stimulate you intellectually?** | | | | |
| Yes | 18 (58.1) | 18 (64.3) | 24 (82.8) | 0.106 |
| No | 13 (41.9) | 10 (35.7) | 5 (17.3) | |
| **Assessment of the level of attention during the scientific communication activity** | | | | |
| Very attentive | 21 (67.7) | 19 (67.9) | 24 (82.8) | 0.334 |
| Not attentive | 10 (32.3) | 9 (32.1) | 5 (17.2) | |
| **AWARENESS** | | | | |
| **Evaluation of the clarity of the information presented** | | | | |
| Very clear | 28 (100) | 22 (100) | 27 (100) | - |
| Not clear at all | 0 | 0 | 0 | |
| **Evaluation of the relevance of the information presented** | | | | |
| Very relevant | 28 (90.3) | 25 (89.3) | 28 (96.5) | 0.067 |
| Not relevant | 3 (9.7) | 3 (10.7) | 1 (3.5) | |
| **ADVOCACY/POLICY INFLUENCE** | | | | |
| **Feel equiped to contribute to the improvement of health services in rural areas** | | | | |
| Equiped to contribute to the improvement of health services in rural areas | 8 (23.5) | 9 (25.0) | 6 (16.2) | 0.620 |
| Not equiped to contribute to the improvement of health services in rural areas | 26 (76.5) | 27 (75.0) | 31 (83.8) | |
| **Motivation to improve health care and services in rural areas** | | | | |
| Motivated to improve health care and services in rural areas | 27 (79.4) | 24 (66.7) | 27 (73.0) | 0.487 |
| Not motivated to improve health care and services in rural areas | 7 (20.6) | 12 (33.3) | 10 (27.0) | |
| **ENJOYMENT** | | | | |
| **What is your overall assessment of the tool?** | | | | |
| Very pleasant | 22 (71.0) | 16 (59.3) | 26 (89.7) | 0.002 |
| Not pleasant at all | 9 (29.0) | 11 (40.7) | 3 (10.3) | |
| **Would you recommend this scientific communication tool to someone else?** | | | | |
| Yes | 21 (67.7) | 13 (46.4) | 26 (89.7) | < 0.001* |
| No | 10 (32.3) | 15 (53.6) | 3 (10.3) | |
| **Did the scientific communication tool move you?** | | | | |
| Yes | 19 (61.3) | 4 (14.3) | 19 (65.5) | < 0.001 |
| No | 12 (38.7) | 24 (85.7) | 10 (34.5) | |

P-Value with * were calculated with Fisher Exact test. Others were calculated with Chi-squared test

further data stratification and sorting based on the qualitative data. This approach helped reinforce and enrich our research outcomes.

## Accessibility

All participants (100%) in the circus show group reported that their intervention can help make scientific research accessible compared to 70,0% for the webinar and 67,9% for the report.

Discussion group participants articulated:

*"Some people might be initially more attracted by the artistic aspect of the presentation, leaving with extra scientific information"* (P74, circus show, decision maker);

*"It made me want to find solutions to the problems presented. It also made me want to go read the actual studies cited."* (P75, circus show, student);

And *"Look, here, we're all from different backgrounds, it touched us all. In any case, I find that it is . . . It has a strength, there, to join people of various sectors, of various concerns. (. . .) There is a concern for rigor while at the same time making it interesting.* (P48, circus show, researcher).

Additionally, when asked in the post-intervention questionnaire: "Do you believe that an artistic production can be a scientific communication tool?", participants from all intervention groups responded positively. As participants reported:

*"Without a doubt, it would be a necessary and indispensable tool to make scientific research accessible and democratic . . .and this, to different age categories and social status and geographical territory and type of intelligence"* (P89, *circus show*, citizen);

*"To go through the emotion, the senses, a well-told story, often leaves a more lasting impression and a deeper understanding of a complex social issue";*

(P49, *report*, citizen);

*Yes, if done rigorously, especially when the aim is to raise audience awareness. (P27, webinar, health professional);*

and: "*I thank you (. . .) it made a difference in my personal life. It's still a study and a research project, but it has made a difference in my life. I imagine that it can make a difference in the lives of many people.* (P51, circus show, citizen)

Participants from in all interventions raised the question of the target audience. The report and webinar were identified as a tool for a selected public, as mentioned by these participants in the report group:

*"The language and the format are still quite hermetic, reserved for a category of academics or people with an interest in scientific research"*

(P48, report, researcher);
and "*It is a communication tool that is not designed to communicate to the largest number of people. It seems to be developed by scientists for scientists*"
(P61, report, artist).
In contrast, the circus show stood out for reaching a wide range of stakeholders:

*"I believe that for the general population it can democratize the scientific articles"* (P34, Webinar, citizen);

*"By popularizing scientific research in this way, anyone, from different spheres of society, can feel challenged and captivated by the information presented. People who have little interest in scientific research can still enjoy watching the show and absorb the scientific information that the show has to offer"* (P76, circus show, student);

and *"I found that there was no line between governance, the citizen, the professional. It was like we were all mobilized in the same way"*
(P47, circus show, citizen).

## Engagement

Participants in the circus show intervention reported notable levels of intellectual stimulation and sustained attention compared to the other two groups during the intervention. In that matter, participants in the show group quoted:

*"It's easier to stay attentive over a longer period of time"* (P98, circus show, citizen);

*"I was able to understand the whole subject without forcing myself to try to find images or examples"* (P99, circus show, student);

*"It gets people's attention, and it's a very subtle translation of knowledge"*

(P51, circus show, researcher);

and finally, *"I was touched. It managed to keep me awake at a time when I was really tired [laughs]" (*P39, circus show, artist)

Participants noted the circus show's

"*combination of various elements was very effective*: narration, inclusion of verbatim, written visual support which potentiate each othe*r*", (P87, circus show, researcher)

and how it "*stimulates the other senses*" (P98, circus show, citizen) and works on "*different levels of language* (body, emotional, intellectual, factual)" (P89, circus show, citizen).
On the other hand, participants in the report and webinar interventions reported that their KT tool required *"A lot of attention from the readers"* (P41, report, citizen), *"a great effort of concentration"* (P43, report, student) *and "intellectual availability"* (P1, webinar, student)"
As one participant shared:

"*For sure, reading an article, then being focused on it for a long time, it can quickly become tedious, even though this is my field*". (P86, report, decision maker)

## Awareness

Participants in all intervention groups recognized the clarity and relevance of the information presented. No significant differences were observed between interventions regarding self-reported knowledge of health issues in rural area or levels of concern about rural healthcare issues. However, participants in the circus show group favorably highlighted how the relationship between emotion and facts smoothed the understanding and retention of information:

*"You come out of there, and you've experienced something. So you didn't just hear the science, you lived it. And I think that's the best way to learn. You know, it's experiential"* (P51, circus show, researcher);

and *"Art has many forms that allow transmitting messages with complex foundations, but with easy interpretation. 1 image/scene is worth a thousand words".* (P99, circus show, student)

## Advocacy and policy influence

There were no differences found among the three interventions pertaining to participants' feelings of being equipped or motivated to change the situation in a rural setting, as indicated by the Likert-scale inquiries. However, in the open-ended questions of the questionnaires and the focus groups, participants who were exposed to the show intervention spontaneously expressed the positive impact the circus show could have on advocacy and policy influence:

*"I found it effective. As a citizen, it makes me want to get involved"*

*(*P80, circus show, manager);

*"I feel really motivated and captivated by this experience"*

(P97, circus show, student)
And "*Even if we're not in reality, it's as if we consider it more important from the outset, and we have more of a desire to mobilize, let's say, than if it were presented to us in a report"* (P39, circus show, artist).
One participant also articulated:

*"Instead of looking for, well: 'Why is this happening? What is happening? Who is the culprit? You know, it's so not fair!', it made you want to find solutions more. It's a big overview but it left room for, like, action"*

(P47, circus show, citizen).
Participants also pointed out the humanizing effect of such a performance:

*"You really humanize it, and then you realize that it gives the problem more weight"* (P39 circus show, artist);

and *"Maybe if I had just read a research paper, well, that it would have stayed just on an intellectual level, then. But to live it with a dramatization like that, well, it's definitely more touching. It gives you an emotional understanding and maybe you'll be more motivated to try to solve the problems that these people are facing"* (P49, circus show, citizen).
Finally, one participant shared:

*"I believe that art can link science and action. One feels more concerned, and one appropriates more naturally the knowledge that one has just acquired"*

(P97, circus show, student).

## Enjoyment

A significant number of participants from the circus show intervention qualified the overall assessment of activity as "very pleasant" (89,7%) compared to the webinar (71%) and the report (59,3%). Also, a significantly greater proportion of participants in the circus show intervention would recommend this scientific communication activity to someone else compared to the two other groups (report 59,3% and webinar 71%). To the question "Did the communication tool move you?" the show group did significantly better than its counterparts (Show 65,5%, webinar 61,3 and report 14,3%). Participants mentioned that it facilitated comprehension and improved the experience of the tool:

"*There is a retention, indeed emotional, that begins to get on board because we live emotions, then we are transported by poetry*" (P39, circus show, artist)

"It gets more people to understand the issues and therefore advocate for them" (P80, circus show, decision maker);

and "*It's a much more powerful tool to do it this way than to do it in a report for managers*" (P49, circus show, citizen).

This emotional dimension was underlined in the other conditions as well. Participants in the webinar appreciated "*the use of real-life cases*" (P28, webinar, health professional) and the "*examples that refer to personal experiences*" (P11, webinar, researcher). Participants mentioned these emotional stories facilitate a deeper understanding and can impact a wider audience. Similarly, participants form the report intervention mentioned the "Use of respondents' quotes was very effective", "making the concepts more human" (P56, report, citizen).

Another dimension that emerges is the notion of learning through play. As participants from the show group mentioned:

"*The idea of having experienced fun and playfulness with a background of reflections on important issues is satisfying*" (P74, circus show, decision maker);

and "*The fun side helps to get interested in the subject*"
(P100, circus show, citizen).

Finally, the aesthetic dimension was noted from participants in all groups. Indeed, participants across all interventions emphasized the significance of clear and captivating visuals as a key factor influencing both enjoyment and comprehension. For the report and the webinar participants appreciated the use of colors, illustrations and sharing of quotes from interviews since it helped bring an emotional dimension.

For its part the

"*well illustrated problematics*" of the show "*created very complete mental images at the end of the presentation*". Visual and artistic illustration of data accentuates the message and further marks the memory*" (P51, circus show, researcher)

## Strengths and weaknesses of the interventions

The joint analyses of the qualitative and quantitative data identified strengths and weaknesses for all three interventions (Fig 6).

**Strengths.** Report: Overall, participants from the report intervention appreciated its clarity, classical format and found it accessible. They described it as inexpensive, complete and detailed, making it a great knowledge translation tool as a reference for an informed public. In their words:

"*It gives a good picture of the situation*" (P 44, report, health professional);

"*A lot of very precise information*" (P59, report, manager);

and "*The references supporting the facts presented in the report are indicated and are therefore easily accessible*" (P60, report, citizen).

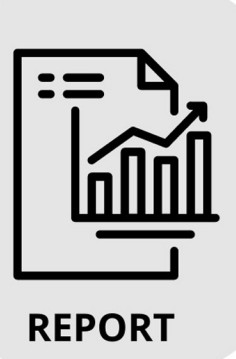

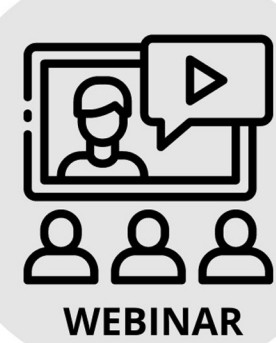

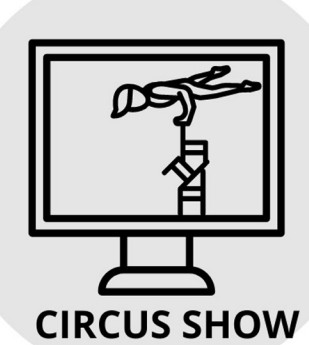

| STRENGTHS | WEAKENESSES |
|-----------|-------------|
| Detailed | Long |
| Complete | Tedious |
| Clear | Boring |
| Classic | Not creative |
| Archivable | For an informed public |
| Inexpensive | |
| For an informed public | |

| STRENGTHS | WEAKENESSES |
|-----------|-------------|
| Constructive | Slow |
| Coherent | Not innovative |
| Trustworthy | For an informed public |
| Readily available | |
| For an informed public | |

| STRENGTHS | WEAKENESSES |
|-----------|-------------|
| Innovative | References not readily available |
| Entertaining | |
| Fun | Rudimentary |
| Easy | |
| Captivating | Work on the integration of the circus |
| Engaging | |
| For a wide audience | |

**Fig 6. Strengths and weaknesses of each intervention.**

Webinar: Participants found the webinar easily available, constructive, coherent and the presence of the primary researcher instilled a sense of trust, reaffirming its status as a valuable online knowledge translation tool for an informed public:

*"To have access to this kind of tool delivered in this way is great. Even though the study is very laborious, I was offered a complete summary, felt, with experience, by someone involved. This is how to get my attention, and above all, when I am talked to and made to feel concerned and that I can even be a positive actor."* (P32, webinar, citizen)

Circus show: Participants in the circus show intervention valued its innovative, entertaining, fun, easy, captivating, and engaging nature, making it a relevant addition to traditional knowledge translation method for a wider public. The following is a list of expressions that participants articulated to describe the circus show:

*"Entertaining, engaging, fantastic, light and fun to watch, beautiful, playful, truly magical, touching, interesting, clever, original, intriguing* and *efficient"*.

Participants spontaneously shared such reactions as:

"*Unique, Incredible*! *The best scientific presentation ever*"

(P87, circus show, researcher)

"*Wow!*"; (P48, circus show, researcher)

"*Loved it*"; (P76, circus show, student)

and "*I have attended the most beautiful conference I have ever attended in my life*! *That was my feeling, as a researcher, when I finished listening to the conference*" (P51, circus show, researcher).

**Weaknesses.**  Report: some described the report as: "*long, tedious, boring, uncreative*, and *dry*".

Webinar: While the majority of comments were positive, a few stated that the webinar was not innovative and slow.

Circus show: Two participants from the show group mentioned difficulty identifying weaknesses, as one mentioned: "*I know that this is the kind of exercise that humans are often asked to do, to say what are the strong points, then the weak points. I really thought about it, really tried to find weaknesses, and I don't have any*" (P47, circus show, citizen). However, some participants mentioned it could be perceived as "*rudimentary*". Others qualified the integration of circus elements as distracting, and some raised the question of cost. As participants from the show mentioned:

"*A show like that requires a lot of logistics*" (P49, circus show, citizen);

"*Maybe more realistic in the form of narrative content like a play than a circus show which personally threw me off. I would watch for example if the person was going to fall out of the pile of boxes rather than listen*" (P71, circus show, health professional);

And "Access to the documents you refer to would be appreciated"

(P99, performance, student).

## Discussion

### Key findings and contribution of the study

Our study aimed to enhance our understanding of arts-based knowledge translation by conducting a comprehensive evaluation and comparison of the circus show with traditional knowledge translation methods. We examined four variables of the arts-based knowledge translation as suggested by Kukkonen: accessibility, engagement, awareness, advocacy/policy influence, and added a fifth variable, enjoyment. Consistent with previous research [48], our results emphasize the significance of integrating the arts into scientific research. This integration not only fosters the creation of innovative communication techniques but also facilitates effective knowledge translation and mobilization [49, 50]. These outcomes are achieved by harnessing the power of emotional [51] and aesthetic [52, 53] mechanisms to engage and captivate audiences.

In relation to our hypothesis, the quantitative data analysis a significant difference between the three interventions on «Accessibility" of the knowledge translation method and the level of "Enjoyment," which was further corroborated by qualitative data. Our findings highlighted the positive influence of an artistic interpretation of a scientific study as an innovative knowledge

translation method targeting the public. While traditional knowledge translation methods were well-received, participants in the circus show intervention unanimously recognized its role in enhancing the accessibility of science. Moreover, they expressed a greater sense of enjoyment with the circus intervention and demonstrated increased willingness to engage in similar activities in the future, as well as being more likely to recommend them.

Although there were no significant differences in the levels of "awareness," "engagement," and "advocacy/policy change" between the two interventions, the qualitative data suggested that the circus show was, at the very least, equally informative and mobilizing compared to the traditional knowledge translation method. However, the circus show had a more profound impact on the participants. The combination of artistry and scientific evidence easily captured their attention, evoked emotional responses, and humanized the information, making it more compelling and memorable.

Through the comparison of the three interventions, we identified strengths and weaknesses for each tool. Our findings strongly advocate for the development of diverse and global knowledge translation methods that incorporate art-based knowledge translation. These results prompted us to raise important questions about the effectiveness of targeting specific audiences when addressing complex social and medical issues involving a diverse range of stakeholders. The circus show emerged as a unifying medium capable of disseminating information and messages across various levels of literacy and experiences, fostering a shared understanding among stakeholders. This shared ground represents a crucial and initial step towards addressing and improving healthcare issues. Our findings highlight the relevance and complementarity of a circus show as an engaging method for researchers to reach large and diverse audiences, making it a powerful means of raising awareness and promoting advocacy on important topics.

## Arts-based knowledge translation challenges: Costs, evaluation, and organization of Interdisciplinarity

Despite the promising results of integrating art in healthcare knowledge translation, significant challenges remain. First, the costs in terms of time and financial resources associated with such initiatives are substantial. Our team dedicated significant effort to develop the circus show, requiring the establishment of new inter-sectorial mechanisms between our academic and artistic institutions, all amidst the challenges posed by a pandemic.

The entire project required about two years of weekly discussions and writing. The process involved an entire week dedicated to prototyping and testing methods [54] in a rural setting that necessitated accommodations for room and board. The final version was recorded after three weeks of intensive studio work at "Les 7 Doigts de la main's" headquarters. These sessions required technicians, artists, and extra logistical staff, and necessitated specialized equipment and the rental of suitable space. The exact total cost is difficult to establish since: 1) the entire show production was made during the pandemic which complicated the process; 2) it required the active participation of our research team (average of 3 people) on site at all stages of production; and 3) several in-kind contributions were provided from the three major partners. An arts-based knowledge translation endeavor requires unique human and technical resources and present various challenges. These include integrating artists into the knowledge translation process, allocating time to translate data into artistic expression and managing the logistics associated with creating and presenting a show. The other two traditional knowledge translation interventions were executed with relative ease, as their costs were already integrated into the overall organization of a conventional scientific research project. For this study, the report was already written, and it was essential for the design of the show. The

PowerPoint presentation was taped in one take by the principal investigator (R. Fleet). While the exact cost of the show intervention is difficult to ascertain, they are beyond those of the traditional knowledge translation methods. However, the same could be said of the impact. We unfortunately did not evaluate the SROI (Social Return on investment), and therefore could not ascertain a correlation between the costs incurred and the medium to long-term impact, including the involvement of the artists, companies, and institutions themselves.

Other challenges included evaluating art within a scientific paradigm and navigating interdisciplinary collaborations across institutions with distinct disciplinary boundaries [55]. In their scoping review Boydell *et al.*, reported that, "arts-based methods, call for a distinct way of dealing with impact, notions of rigor and ways of evaluating assertions or not" [56]. The questions of how to capture art experiences through scientific inquiry, the academic legitimacy of such endeavors [57] and how artistic research translates into non artistic domains [19] are addressed in arts-based knowledge translation literature and were encountered in our study. Similarly, evaluating the social impact requires the adoption of novel performance indicators capable of capturing sustainable, adaptable, and complex social change. Most importantly, we consider these challenges as pivotal opportunities to maximize the impact of health research, leverage return on investment and foster sustainable change by uniting allies from various fields and paradigms. In this regard, it is noteworthy that since conducting this research, the circus show has been presented over 30 times to professionals and institutions in the artistic, healthcare and research domains and was received with great interest. Furthermore, since this study, CRITAC has developed three new interdisciplinary studies on arts-based knowledge translation.

## Strengths and limitations

To the best of our knowledge, this study stands as the first to compare arts-based knowledge translation with traditional knowledge translation methods. However, it is crucial to acknowledge and address several limitations that were encountered:1- The sample size for the randomized study was relatively small, which might affect the generalizability of the results; 2- No initial power calculation was conducted before the study, which could have provided insights into the appropriate sample size and statistical significance; 3- The questionnaires used in the study were developed in-house and not validated, which may have impacted the reliability and validity of the data.4- Considerable resources were dedicated to the positively valued KT method, i.e., the circus show, which may have introduced an unintended advantage, potentially skewing the results; 5-The complex nature of the circus show intervention might limit the generalizability of the findings to other settings or populations.

Moreover, it is essential to acknowledge that our inclusion criteria may have inadvertently excluded certain segments of the population of interest. For instance, individuals from rural areas or marginalized groups without access to the internet or technology, or those who do not speak French, may not have been adequately represented in the study. These limitations should be considered when interpreting the results and considering the broader implications of the findings.

Our study however possesses several notable strengths. Being the first of its kind to compare arts-based knowledge translation with traditional knowledge translation methods, we embarked on this investigation without prior knowledge of effect sizes, precluding the use of power calculations. Nonetheless, both quantitative and qualitative findings consistently supported our initial hypotheses, indicating a robust and reliable trend. Moreover, our sample size of 88 participants aligns well with similar studies present in the arts-based knowledge translation literature [23]. Although arts-based knowledge translation initiatives and research are

gaining momentum, there remains a lack of consensus regarding the taxonomy or terminology for these interventions. Additionally, the absence of adequate interdisciplinary metrics and validated questionnaires at the time of our study presented a challenge. Despite these hurdles, our research is pioneering innovative approaches to knowledge translation and arts-based knowledge translation, paving the way for further advancements in this field.

Furthermore, we took significant measures to recruit and randomize participants. We employed email invitations and boosted social media posts from all three partners consistently over a three-week recruiting period. The Research Chair's social media statistics indicate that our outreach potentially reached 8000 individuals. However, recruiting participants proved challenging, especially during the pandemic, as it required a substantial time commitment of approximately 3 hours for their involvement in the research. Given these constraints and the limited timeframe for recruitment, we doubt that we could have recruited more subjects, regardless of any power calculations.

Additionally, our recruitment efforts were bound by methodological and ethical considerations. For instance, we refrained from promoting the circus show, even if it was produced by an internationally acclaimed circus company, to prevent biasing subjects towards any intervention. This decision raises an interesting unresolved question: If we had briefly described the three interventions and their purposes and allowed subjects to choose their interventions, which one would they have opted for? This highlights the importance of methodological rigor in ensuring unbiased participant selection and adds to the complexity of interpreting the study's outcomes.

The hybrid nature of this study generates great methodological, financial, and publishing challenges. However, it has extensive innovative value and potential. In its essence, it is pragmatic and challenges traditional knowledge translation methods and evaluations. Its intermediary state paves the way for new perspectives, questions, and methods to tackle complex issues. By analyzing participants' responses to open-ended questions from the questionnaires and discussions in the focus groups, we were able to gather valuable recommendations about knowledge translation method. Health professionals, researchers, students, decision makers, artists and members of the general public reported a need to develop a comprehensive knowledge translation strategy that encompass a diverse range of methods. This approach aims to address the varying competencies and interest of stakeholders. The participants also emphasized the importance of ensuring the accessibility and availability of scientific research through innovative and interactive knowledge translation methods. They highlighted the importance of using knowledge translation approaches to provide insight into solutions, explore possible courses of action and actively engage a wide range of individuals including the researcher themselves. Finally, the opportunity of conducting this comparative study with three interventions, contributed to raising awareness and dialogue around access to rural healthcare services. If nothing else, a public from diverse backgrounds now has a deeper understanding of the challenges of providing equitable access to emergency care in rural areas.

## Further directions

Arts-based knowledge translation is an ever-expanding and diverse domain, encompassing studies with various designs, sample sizes, populations, themes, and outcomes [20, 26, 58, 59]. Alongside this growth, comprehensive design, evaluation, and implementation models are being developed [24, 60, 61]. Moving forward, we recommend conducting further research that explores awareness, engagement, advocacy/policy influence, and enjoyment. Additionally, investigating the underlying mechanisms at play in arts-based knowledge translation, particularly its dimensions of learning through emotions, aesthetics, and play, would be valuable.

Moreover, assessing the retention of information and the extent to which participants take concrete action towards change in the 6–12 months following the interventions can provide insights into the lasting impact of arts-based knowledge translation methods.

We recommend investigating even simple arts-based knowledge translation strategies to improve impact. For example, "enhanced" PowerPoint presentations with better delivery, animations, visuals, and sound, may even improve knowledge translation. Yet, whether it qualifies as arts-based knowledge translation depends on how effectively it leverages artistic and scientific approaches to engage and communicate with the audience. Moreover, it is not necessary to partner with an internationally acclaimed circus company; local, community, or school-based arts groups can collaborate with researchers to co-develop knowledge translation acts. The imperative is to foster collaborative and participative co-development of innovative solutions that integrate both artistic and scientific approaches to address public healthcare issues.

We encourage future research teams to:

1. Develop arts-based knowledge translations strategies that involve academic research, decision-makers, healthcare professionals, managers, artists, and the public in co-creating, implementing, and evaluating arts-based knowledge translation methods focused on social innovation and lasting impact.

2. Examine the key facilitators, barriers, and good practices in the process of co-creating an arts-based knowledge translation object.

3. Develop and investigate adaptive, global, and agile models for evaluating and organizing well-founded information with social, emotional, interactive, bodily, and aesthetic dimensions specific to arts-based knowledge translation strategies within academic, decision-making, artistic, and associated institutions.

Despite the constraints of presenting a virtual recording of the circus show, it remains a pertinent question how a live show would compare. Circus as a live medium thrives on inclusivity, participation, and interactivity, which are vital aspects for optimal impact. Future studies should explore this dimension.

Other relevant questions to investigate include:

1. How to generate initial interest in a given knowledge translation / arts-based knowledge translation, which is often overlooked in most KT process models.

2. To what extent and why an arts-based knowledge translation object can influence decision-making, leading to the implementation of recommended actions to solve specific problems studied in a research project.

Finally, although our study introduced circus as an arts-based knowledge translation for investigation, it did not directly explore the specific effect of circus arts as an arts-based knowledge translation. Nevertheless, participants did mention the magical dimension, movement, and tension attributed to circus. Further research into circus as arts-based knowledge translation and its intersection with medicine is warranted.

## Conclusion

Given that traditional knowledge translation strategies often take more than a decade to achieve impact, exploring alternative approaches becomes essential [62–65]. Our study is pioneering as the first to compare an arts-based knowledge translation method (circus show) with traditional knowledge translation interventions like webinars and scientific reports. The arts-based knowledge translation object stands out in attracting, informing, engaging, and

mobilizing audiences through emotion, aesthetics, and play, outperforming traditional knowledge translation interventions on most measures. We provide crucial insights into the purposes and limitations of each approach, emphasizing the need to develop global, innovative, and diverse knowledge translation methods and strategies that involve stakeholders, decision-makers, practitioners, citizens, and artists in a co-creative approach.

## Supporting information

**S1 Checklist. Human participants research checklist.**
(DOCX)

## Acknowledgments

We wish to thank circus artists Anne-Fay Johnston-Audet, Robert Cookson, Charlotte Fallu, Alexandre Seim, Josiane Lamoureux, Élise Legrand, Simon Durocher-Gosselin and graphic designer and animator Mathieu Fortin for their creative input throughout the project. We also wish to thank Emanuel Bochud, Marion Cossin, Marie-Pier Parent, Nicolas Scherrer and Laurie-Anne Bergeron-Drolet for their contribution in data collection. We wish to acknowledge Planning, programming and research officer (epidemiology and biostatistics) Stéphane Turcotte for his support in data analysis. This study is part of first author Julie Théberge's PhD program, Laval University, supervised by Richard Fleet (PI) and Jocelyn Robert.

## Author Contributions

**Conceptualization:** Julie Théberge, Catherine Turgeon-Pelchat, Patrice Aubertin, Hassane Alami, Richard Fleet.

**Data curation:** Julie Théberge.

**Formal analysis:** Julie Théberge, Mélanie Ann Smithman, Catherine Turgeon-Pelchat, Fatoumata Korika Tounkara.

**Funding acquisition:** Julie Théberge, Patrice Aubertin, Patrick Léonard, Richard Fleet.

**Investigation:** Julie Théberge.

**Methodology:** Julie Théberge, Véronique Richard, Patrice Aubertin.

**Project administration:** Julie Théberge, Patrice Aubertin, Richard Fleet.

**Resources:** Julie Théberge, Richard Fleet.

**Software:** Julie Théberge.

**Supervision:** Julie Théberge, Patrice Aubertin, Richard Fleet.

**Validation:** Julie Théberge, Mélanie Ann Smithman, Catherine Turgeon-Pelchat, Véronique Richard, Patrice Aubertin, Richard Fleet.

**Visualization:** Julie Théberge.

**Writing – original draft:** Julie Théberge.

**Writing – review & editing:** Julie Théberge, Mélanie Ann Smithman, Catherine Turgeon-Pelchat, Fatoumata Korika Tounkara, Véronique Richard, Patrice Aubertin, Patrick Léonard, Hassane Alami, Diane Singhroy, Richard Fleet.

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
