## [Decision Letter · Decision Letter 0]

19 Jan 2024

PONE-D-23-36693Through the Big Top: An Exploratory Study of Circus-Based Artistic Knowledge Translation in Rural Healthcare Services, Québec, CanadaPLOS ONE

Dear Dr. Fleet,

Thank you for submitting your manuscript to PLOS ONE. After careful consideration, we feel that it has merit but does not fully meet PLOS ONE’s publication criteria as it currently stands. Therefore, we invite you to submit a revised version of the manuscript that addresses the points raised during the review process.

We look forward to receiving your revised manuscript.

Kind regards,

Lóránt Dénes Dávid, PhD

Academic Editor

PLOS ONE

Journal Requirements:

"NO authors have competing interests"

5. Please provide a complete Data Availability Statement in the submission form, ensuring you include all necessary access information or a reason for why you are unable to make your data freely accessible. If your research concerns only data provided within your submission, please write "All data are in the manuscript and/or supporting information files" as your Data Availability Statement.

7. We note that Image 2 and 3 in your submission contain copyrighted images. All PLOS content is published under the Creative Commons Attribution License (CC BY 4.0), which means that the manuscript, images, and Supporting Information files will be freely available online, and any third party is permitted to access, download, copy, distribute, and use these materials in any way, even commercially, with proper attribution. For more information, see our copyright guidelines: http://journals.plos.org/plosone/s/licenses-and-copyright.

a. You may seek permission from the original copyright holder of Image 2 and 3 to publish the content specifically under the CC BY 4.0 license. 

8. Please ensure that you refer to Figures 1-5 in your text as, if accepted, production will need this reference to link the reader to the figure.

9. We note that Figure 4 includes an image of a [patient / participant / in the study]. 

If you are unable to obtain consent from the subject of the photograph, you will need to remove the figure and any other textual identifying information or case descriptions for this individual."

Reviewers' comments:

Reviewer's Responses to Questions

**Comments to the Author**

1. Is the manuscript technically sound, and do the data support the conclusions?

Reviewer #1: Yes

Reviewer #2: Yes

2. Has the statistical analysis been performed appropriately and rigorously? 

Reviewer #1: N/A

Reviewer #2: Yes

3. Have the authors made all data underlying the findings in their manuscript fully available?

Reviewer #1: No

Reviewer #2: Yes

4. Is the manuscript presented in an intelligible fashion and written in standard English?

Reviewer #1: No

Reviewer #2: Yes

5. Review Comments to the Author

Reviewer #1: The article on knowledge translation is an interesting read, and I rarely come across this research method. After careful review, I have the following concerns that I hope the author can address before the second round of review:

1. I noticed a quoted statement before the introduction section, but I'm unsure of its purpose and intended effect. It would be helpful to clarify why this statement is included and what it aims to achieve.

2. Knowledge translation as a method is not widely known, and I suggest that the author provides a more detailed introduction to it in the first paragraph. Currently, I did not receive a clear and concise explanation of knowledge translation from the beginning of the introduction.

3. I encourage the use of abbreviations in academic papers to avoid excessive and complex terminology. However, it should not hinder the reading and understanding of the paper. "Knowledge translation" is not a lengthy or difficult-to-understand phrase, so I recommend reconsidering the use of "KT" as an abbreviation. The frequent appearance of "KT" reminds me that I need to spend time understanding what it stands for.

4. "Objective" and "Hypothesis" should not be subheadings in the introduction section. Please integrate them directly into the introduction.

5. It might be beneficial to include a flowchart in the "Procedures" subsection.

6. Many of the English expressions in this manuscript do not adhere to academic standards, such as the use of "Image" instead of "figure." Similar difficulties in understanding are also reflected in sentences, headings, and the structure of the manuscript. I recommend that the author read some other articles already published in this journal for reference. Please refer to: https://doi.org/10.3390/su15031948

https://doi.org/10.1108/IJSHE-09-2018-0150

Good luck.

Reviewer #2: The study examines a current topic, the objectives are clear. To evaluate and compare the impact of an

ABKT intervention with two traditional KT interventions is quite a new topic. I recommend doing a literature review of the investigation in a separate chapter. To explore the application possibilities of knowledge management, I recommend using the following literature:

Ogutu H. et al (2023). Theoretical Nexus of Knowledge Management and Tourism Business Enterprise Competitiveness: An Integrated Overview

The methods used are appropriate, random sample selection in research is usually not lucky, but in this case it is justified.

The obtained results and their presentation are acceptable, but at the same time, I recommend a more visual presentation of the results using figures.

I recommend the study for publication.

6. PLOS authors have the option to publish the peer review history of their article (what does this mean?). If published, this will include your full peer review and any attached files.

Reviewer #1: No

Reviewer #2: No

---

## [Author Response · Author response to Decision Letter 0]

21 Mar 2024

Dear members of the revision committee, 

We would like to express our gratitude to the editors and reviewers for their valuable time and insightful comments on the manuscript we submitted. Below, we provide responses to each comment made by the editors and reviewers and detail the corresponding modifications made to the manuscript. We appreciate this collaborative process, which enhances the quality of our manuscript and moves us forward in our goal of submitting to PlosOne.

Journal Requirements:

1.Ensure that your manuscript meets PLOS ONE's style requirements. Changes were made according to guidelines: the label “Image” was changed to “Figure,” figure file names and citations were changed to fit requirements, changes were made to headings, blank lines were deleted in tables 1 and 2.

 Editors’ revision:

1-2. Depositing protocol and data in a repository. Thank you for the information, this will be considered. 

3.‘Funding Information’ and ‘Financial Disclosure’ sections. After verification, it appears the grant numbers are correct. However, in the submission form a change was made from “Fonds de Recherche du Québec-Nature et Technologies” to “Fonds de Recherche du Québec- Société et Culture” (Quebec Research Funds- Nature and Technologies to Society and Culture). The funding information was deleted from the manuscript as advised. Also, an updated financial disclosure statement was added in the cover letter in the fourth paragraph of a new version of the cover letter. In the new version of the cover letter, the opening sentence was modified to “Please accept the enclosed REVISED manuscript entitled “Through the Big Top: An Exploratory Study of Circus-Based Artistic Knowledge Translation in Rural Healthcare Services, Québec, Canada" for consideration for publication as an original article in Plos One.”

4. Competing interest section. The sentence: «The authors have declared that no competing interests” was included in the last paragraph of the cover letter.

5. Data Availability Statement in the submission form. The article includes all relevant data except for the French verbatim transcripts, which can be obtained upon request. Therefore, the statement "All data are in the manuscript and/or supporting information files" was revised to the Data Availability Statement.

6. Validated ORCID iD. Completed with submission.

7. Copyrighted images for “Image” 2 and 3, now labelled “Figure” 2 and 3 in the new manuscript. The authors are the owner of the material. The following text was included in the manuscript in this format for pages 9, 12, 13, 14, 15 and 28 :“Reprinted from [ref] under a CC BY license, with permission from [name of publisher], original copyright [original copyright year].” 

 8. Change label Images for Figures. Completed in text and figures legends for figures 1 to 5 , and figure legend only for figure 6. Pages 9, 12, 13, 14, 15, 28.

9. Consent Figure 4. Figure 4 features artists from the circus show intervention, not patients or participants in the study. The broadcasting of the show and images is stipulated in their contract. Since they are not study participants, the methods section and ethics statement remained unchanged. 

10. Review reference list. The review was completed. No changes were made to the list. 

Reviewers' Comments

Provide data as part of the manuscript or its supporting information. Please refer to previous response regarding comment #5 in the editor's comments section.

Reviewer #1: 

1. Quoted statement before the introduction section. Opening quotes can be a way of setting the table to what the article will be addressing, in a more poetic or philosophical approach. After reflection, we deleted the quote since it can bring confusion.

2. More detailed introduction to knowledge translation as a method. Completed, page 4. In the first paragraph of the introduction, the definition of knowledge translation is provided as follows: “In the context of Canadian healthcare, knowledge translation is defined as a dynamic and iterative process that includes synthesis, dissemination, exchange and ethically-sound application of knowledge to provide Canadians more effective healthcare services and products and strengthen the healthcare system”. To enhance clarity, the following sentence was included in the manuscript: “It aims to move important health research evidence into the hands of people and organizations who can put it into practice.”

3. Reconsider the use of "KT" as an abbreviation for "Knowledge translation". 

To enhance readability, we replaced the abbreviations “KT” with “knowledge translation” and “ABKT” with “arts-based research translation” throughout the text.

4. Integrate "Objective" and "Hypothesis" sections directly into the introduction. Completed, page 6.

5. Include a flowchart in the "Procedures" subsection. To provide clarity, the legend of the figure labelled “Methodology Timeline” (Figure 5) on page 15 was updated to “Procedure Flowchart.'"

6.English expressions and academic standards. We revised the entire text to conform with academic standards and adhere to the publishing guidelines of PlosOne. This involved rectifying terminology such as "images" and "figures," as well as refining section headings and overall formatting. Our goal was to ensure that the language used was appropriate for an interdisciplinary project merging aspects of circus arts with healthcare research.

Reviewer #2: 

Literature review of the investigation in a separate chapter. Thank you for the comment and the reference. Indeed, exploring the application possibilities of knowledge management is an interesting avenue for such projects. A literature review was recently submitted as part of first author’s PhD project. In the case of the manuscript submitted here, special intention was paid to incorporate relevant references. While the 65 references included in the manuscript may not constitute a comprehensive literature review, we believe they cover a wide range of topics related to knowledge translation and arts-based knowledge translation research.

Visual presentation of the results using figures. The comment is very relevant. We are indeed developing additional visual, creative and participatory tools to disseminate the results of this research. Regarding this manuscript, after discussion with our co-authors, we opted to present the results in a classic format using words and tables instead of visual figures. This decision aligns with our goal of establishing a foundational reference point.

We wish to thank you again for the comments, encouragement, and support. 

We look forward for the next steps.

Richard Fleet

---

## [Editor Report · Decision Letter 1]

27 Mar 2024

Through the Big Top: An Exploratory Study of Circus-Based Artistic Knowledge Translation in Rural Healthcare Services, Québec, Canada

PONE-D-23-36693R1

Dear Dr. Richard Fleet,

We’re pleased to inform you that your manuscript has been judged scientifically suitable for publication and will be formally accepted for publication once it meets all outstanding technical requirements.

Kind regards,

Lóránt Dénes Dávid, PhD

Academic Editor

PLOS ONE
---

## [Editor Report · Acceptance letter]

5 Apr 2024

PONE-D-23-36693R1 

PLOS ONE

Dear Dr. Fleet, 

I'm pleased to inform you that your manuscript has been deemed suitable for publication in PLOS ONE. Congratulations! Your manuscript is now being handed over to our production team.

Kind regards, 

on behalf of

Dr. Lóránt Dénes Dávid 

Academic Editor

PLOS ONE